# Preparation of Ibuprofen-Loaded Inhalable γCD-MOFs by Freeze-Drying Using the QbD Approach

**DOI:** 10.3390/pharmaceutics16111361

**Published:** 2024-10-24

**Authors:** Anett Motzwickler-Németh, Petra Party, Péter Simon, Milena Sorrenti, Rita Ambrus, Ildikó Csóka

**Affiliations:** 1Faculty of Pharmacy, Institute of Pharmaceutical Technology and Regulatory Affairs, University of Szeged, 6720 Szeged, Hungary; nemeth.anett@szte.hu (A.M.-N.); party.petra@szte.hu (P.P.); csoka.ildiko@szte.hu (I.C.); 2Faculty of Pharmacy, Institute of Pharmaceutical Chemistry, University of Szeged, 6720 Szeged, Hungary; simon.peter.03@szte.hu; 3Department of Drug Sciences, University of Pavia, 27100 Pavia, Italy; milena.sorrenti@unipv.it

**Keywords:** ibuprofen, γ-cyclodextrin metal-organic framework, freeze-drying, targeted delivery, alternative administration route, dry powder inhaler, inclusion technology, drug delivery

## Abstract

Background/Objectives: Research on cyclodextrin-based metal-organic frameworks (CD-MOFs) is still in its infancy, but their potential for use in drug delivery—expressly in the lung—seems promising. We aimed to use the freeze-drying method to create a novel approach for preparing CD-MOFs. MOFs consisting of γ-cyclodextrin (γCD) and potassium cations (K^+^) were employed to encapsulate the poorly water-soluble model drug Ibuprofen (IBU) for the treatment of cystic fibrosis (CF). Methods: Using the LeanQbD^®^ software (v2022), we designed the experiments based on the Quality by Design (QbD) concept. According to QbD, we identified the three most critical factors, which were the molar ratio of the IBU to the γCD, incubation time, and the percentage of the organic solvent. light-, scanning electron microscope (SEM) and laser diffraction were utilized to observe the morphology and particle size of the samples. In addition, the products were characterized by Differential Scanning Calorimetry (DSC), X-ray Powder Diffraction (XRPD), Fourier Transform Infrared Spectroscopy (FT-IR) and nuclear magnetic resonance spectroscopy (NMR). Results: Based on characterizations, we concluded that a γCD-MOF/IBU complex was also formed using the freeze-drying method. Using formulations with optimal aerodynamic properties, we achieved 38.10 ± 5.06 and 47.18 ± 4.18 Fine Particle Fraction% (FPF%) based on the Andersen Cascade Impactor measurement. With these formulations, we achieved a fast dissolution profile and increased IBU solubility. Conclusions: This research successfully demonstrates the innovative use of freeze-drying to produce γCD-MOFs for inhalable IBU delivery. The method enabled to modify the particle size, which was crucial for successful pulmonary intake, emphasizing the need for further investigation of these formulations as effective delivery systems.

## 1. Introduction

Ibuprofen (IBU), a propionic acid derivative (Figure 1), is a classical Biopharmaceutical Classification System (BCS) class II nonsteroidal anti-inflammatory drug (NSAID) that is mostly administered orally to relieve acute pain and inflammation. Its effect, like other NSAIDs, is based on its ability to inhibit the production of prostaglandins by reducing the activity of cyclooxygenase enzymes (COX-1 and COX-2) [1]. IBU is extremely poorly soluble in aqueous media [2] as its solubility is 0.021 mg·mL^−1^ [3]. Therefore, the rate of dissolution from the solid dosage forms that are currently available is limited. The required dosage for a therapeutic effect in an adult is approximately 20–30 mg, whereas the provided dose is more than 10 times this amount [4] because of its poor bioavailability and first-pass metabolism. In addition to being economically inefficient, this significantly increases the occurrence of side effects caused by IBU [5], e.g., stomachache, indigestion, or gastrointestinal bleeding and kidney injury [6]. To overcome this issue, alternative administration routes have been investigated for the delivery of IBU, including ocular [7], nasal [8], inhalation [9,10], and transdermal routes [11]. The use of these routes offers the opportunity to reduce the amount of active substances by bypassing the first-pass mechanism and exerting their effect locally.

Although IBU is not primarily used for cystic fibrosis (CF) therapy, it is the only NSAID approved for chronic use in CF, which is an autosomal recessive inheritance. Among the symptoms, in addition to inflammation and infection, the production of excessive viscous mucus can be observed. This mucus overproduction arises from the dysregulated transport of ions and water through the epithelial cells [6]. Consequently, the impaired epithelial function leads to inadequate oxygenation of the lungs. Since inflammation is a characteristic of the lung disease of CF and contributes to lung destruction, anti-inflammatory therapy may be helpful in slowing disease progression [5]. Treatment of the disease is especially important in the early stages [12,13,14], because in the case of CF patients, abnormal mucus production usually begins immediately after birth [15], which later leads to infection and inflammation, which in the end causes death. Several clinical trials document the benefits and relative safety of long-term use of high-dose IBU in CF patients [16,17,18]. We therefore declare that IBU may be useful in the treatment of CF, but the challenge lies in the fact that, in addition to having very poor water solubility, due to the mentioned side effects, it is necessary to introduce IBU through an alternative administration route, which is evident in the case of CF: the lung. With local therapy, pulmonary drug delivery can greatly improve drug absorption due to the unique physiological characteristics of the lung, such as a large alveolar surface area [19], extensive vascularization, and thin alveolar epithelium. However, since there is just 5.6 mL/kg extravascular lung water in normal lungs [20] to keep the tissues moist, it is advisable to increase the solubility of poorly water-soluble active ingredients such as IBU in order to achieve the desired absorption. It is noteworthy to mention that patients with cystic fibrosis (CF) exhibit significantly higher lung water content [21,22], which can be attributed to several factors such as edema and infections. Utilizing cyclodextrins (CDs), the aforementioned issues can be resolved.

CDs are truncated cone-shaped cyclic oligosaccharide compounds which are formed by six (αCD), seven (βCD), or eight (γCD) *D*-glucopyranose units, bound via α-1,4- glycosidic linkages (Figure 2a). The inner cavities of CDs are formed by the hydrophobic carbon backbones of glucopyranose monomers, making the interior hydrophobic (Figure 2b) [23]. Due to this structural feature, the use of CDs is mostly focused on their use as solubilizers for molecules that are poorly soluble in water [24,25,26,27]. In addition to absorbing poorly water-soluble compounds, CDs can also modify the physicochemical characteristics of these molecules; for example, they can increase stability [24,25,26,27,28] and permeability [28,29]. The spread of CDs in the food industry and the pharmaceutical industry is made possible by the fact that they are edible and not toxic [30] for human use. With the application of CDs, the listed positive benefits provide the opportunity to increase IBU’s solubility in aqueous media and, consequently, its bioavailability while utilizing less of the active ingredient. In addition to the water solubility of the active ingredient, there are other factors that strongly influence the effectiveness of dry powder inhalers (DPIs), mainly aerodynamic particle size, particle shape, texture, and density [31]. For the use of CDs in the lungs, more and more research is appearing in the literature on the use of cyclodextrin-based metal-organic frameworks (CD-MOFs) [32,33,34,35,36] as they have excellent aerodynamic properties.

γCD-MOFs have been studied more than any other type of CDs thanks to their largest specific surface area (approximately 1220 m^2^/g [37]), richest pore channels (spherical pore diameter is about 0.4–1.7 nm [37]), structural symmetry, and stability. For γCD-MOFs, six γCD organic ligand molecules are bridged by alkali metal salt (mostly potassium ions (K^+^)) through coordination bonds to form a repeating unit (γ-CD)_6_, which is bridged by K^+^ and extended along the a, b, and c crystallographic axes in a specific manner [38], forming a new topological structure (Figure 2c). Because of their uniform nanoporous structure, CD-MOFs demonstrated greater aerosolization performance when compared to commercial inhalation carriers [39]. The biocompatibility of CD-MOF in pulmonary administration has also been proven. Yong et al. demonstrated that CD-MOFs can be directly used as a DPI without any other excipients [33]. According to their research, the aerosolization of the carriers depended primarily on the particle size and not on the flowability, so the added excipient did not promote the deep deposition of the sample in the lungs. Another study on pulmonary applicability of CD-MOFs reported that curcumin was loaded into CD-MOFs and compared with micronized curcumin (Cur-CD-MOF) [32]. The results showed that the Cur-CD-MOF, in addition to exhibiting excellent aerodynamic performance, showed a much faster drug release rate than that of micronized curcumin, while CD-MOFs could dramatically improve the solubility of various hydrophobic active pharmaceutical ingredients (APIs). Even if a lot of studies on the pulmonary application of CD-MOFs have been published, the quantity of these studies is still quite low. Nonetheless, they appear extremely promising, and an increasing amount of research is required to explore and comprehend the topic more thoroughly.

Vapor diffusion is the most widely used method for synthesizing CD-MOFs; however, this process can take 3–5 days or weeks, requires the use of a large amount of organic solvent and provides low product yield [40]. Furthermore, in several cases, the active ingredient was loaded in a separate step [41,42,43], which means a multi-step process. Consequently, the development of alternative protocols for CD-MOF production was deemed essential, with proposed advantages including enhanced product yield, reduced preparation time, and minimized amount of organic solvent. Alternative production techniques already exist in the literature [38], such as microwave-, ultrasound-, or modulator-assisted synthesis. The use of modulators further complicated the production, as they were often inseparable from the final product [44]. In addition, the previously mentioned techniques require a subsequent drying process. Despite the freeze-drying method occasionally mentioned in literature as a final step of synthesis [42,45,46,47], there has been limited comprehensive investigation into its suitability for complex preparation from solutions, even though analogous experiments have been carried out using the spray-drying method [36,48]. In comparison to the traditional vapor diffusion method, spray-drying presents a promising alternative for production, offering advantages such as higher product yield, reduced synthesis time, and tunable particle properties, all achieved in a single step without the need for a subsequent drying process. However, the technique has certain limitations; for instance, its application is not advantageous for heat-sensitive, highly viscous, or highly foaming materials. The alternative freeze-drying technique can provide a solution to the problem. In comparison to spray-drying, freeze-drying is a more time-consuming, complex, and expensive process. However, it retains the advantages associated with spray-drying over the traditional vapor diffusion method. Moreover, freeze-drying can address specific challenges that arise during spray drying and provides an opportunity to assist in the production of a stable product. Taking these factors into consideration, we strongly believe that developing a freeze-drying production process for CD-MOFs could represent a valuable strategy.

Quality by Design (QbD) is a comprehensive methodology in pharmaceutical development that prioritizes systematic, scientific, risk-based, and proactive approaches [49,50]. It emphasizes establishing predefined quality objectives, thorough understanding of product and process, and stringent process control. QbD integrates quality risk management into every stage of formulation, focusing on ensuring consistent product quality. The QbD strategy, outlined in the International Council for Harmonisation of Technical Requirements for Pharmaceuticals for Human Use (ICH) quality guidelines, including Q8 (Pharmaceutical Development [51]), Q9 (Quality Risk Management [52], Q10 (Pharmaceutical Quality System [53]), and Q11 (Development and Manufacture of Drug Substances [54]) aligns with the standards and expectations of both national and international regulatory bodies. In the QbD approach, the first step involves determining the Quality Profile of the Target Product (QTPP) [55], which outlines the essential characteristics crucial for its therapeutic application. After determining the QTPP, it is necessary to define Critical Quality Attributes (CQAs), which encompass the physical, chemical, and microbiological properties directly impacting the safety and efficacy of the product. Following CQAs establishment, it is necessary to identify Critical Material Attributes (CMAs) and Critical Process Parameters (CPPs). CMAs express which Material Properties (MAs) are critical relating to CQAs, while CPPs outline the process parameters that significantly influence the production process. Risk Assessment (RA) plays a key role in this process by evaluating the severity and likelihood of occurrence of identified risks. This assessment guides the prioritization of critical parameters influencing CQAs. Once the critical parameters are identified, factorial experimental designs, such as the 3^3^ Box–Behnken experimental design, are employed [56,57]. These designs allow for the systematic exploration of multiple factors and their interactions, facilitating the optimization of the production process.

In this study, γCD-MOF composed of γCD and K^+^ ions were utilized to form an inclusion complex with the model drug IBU. The Box–Behnken factorial design has been applied based on a QbD approach to produce new formulations. Although the use of QbD is becoming more and more widespread in drug research, we did not find any publication that applied it in the field of CD-MOFs. The preparation process for the approach was freeze-drying. First analyses were conducted to evaluate morphology, particle size, and crystallinity, ensuring the suitability of the particles for targeted lung delivery. Next, investigations focused on the thermostability of IBU and its interaction with γCD-MOF. To evaluate the practical application of the prepared samples, in vitro pulmonary deposition studies were carried out utilizing the Andersen Cascade Impactor. In vitro release assays were also carried out to measure the encapsulated drug’s release profile. These investigations shed light on the potential of γ-CD-MOF complexes as promising carriers for pulmonary drug delivery systems, underscoring the significance of the QbD approach in the field of pharmaceutical sciences.

## 2. Materials and Methods

### 2.1. Materials

IBU was obtained from Sanofi (Veresegyház, Hungary). Other chemicals used in the experiments were γCD (Cyclolab, Budapest, Hungary) and potassium-hydroxide (AnalaR NORMAPUR, Leuven, Belgium). Ethanol 96% (EtOH) (Molar Chemicals Kft, Halásztelek, Hungary), methanol (MeOH) (AnalaR NORMAPUR, Leuven, Belgium), and purified water were used as solvents. Simulated lung fluid contained 0.68 g/L NaCl, 2.27 g/L NaHCO_3_, 0.02 g/L CaCl_2_, 0.1391 g/L NaH_2_PO_4_, 0.37 g/L glycine, and 5.56 mL/L of 0.1 M H_2_SO_4_. All reagents and solvents were used without further purification. For high-pressure chromatography, we utilized a phosphate buffer solution (PBS). This buffer solution was prepared using highly purified water obtained through deionization and filtration using a Millipore purification apparatus, achieving a conductivity of 18.2 MΩ·cm at 25 °C. Additionally, we employed HPLC-grade acetonitrile (LGC Promochem, Wesel, Germany).

### 2.2. Methods

#### 2.2.1. Risk Assessment and Determination of Critical Parameters

The first step was to determine the QTPP based on ICH guideline Q8. Table 1 shows the requirements we have defined in terms of QTPP for IBU-containing pulmonary preparations. We also defined CQAs, CPPs, and CMAs as follows based on the information collected during the preliminary knowledge space development. An Ishikawa cause-effect diagram (Figure 3) was used to summarize the variables affecting the final result. We specified the CMAs and CPPs for achieving the QTPP by highlighting the critical factors from the Material Attributes and Process Parameters. After determining each parameter, the LeanQbD^®^ software (v2022, QbD Works LLC, Fremont, CA, USA) was used to determine their impact on the quality of the target product. In order to accomplish this, we assessed the impact of CPPs and CMAs on CQAs using a three-level scale provided by the software: “low”, “medium”, or “high”. We assigned a severity value of 1 to the low-risk parameter, 3 to the medium category, and 9 to the high category.

#### 2.2.2. Preparation of the Samples

##### Vapor Diffusion Method

CD-MOFs were synthesized using a conventional vapor diffusion method similarly as described in the literature [38]. This method was chosen as the standard protocol for preparing control samples. Based on this, an excess of MeOH was slowly diffused into a beaker that contained the previously mixed solution of γCD (20 mL, 35 mM) and potassium hydroxide (1:8 molar ratio). The procedure was carried out in a closed space at room temperature. After 7 days, the cubic crystals that appeared in the beaker were collected and separated from the remaining solution by vacuum filtration. Subsequently, the cubic crystals underwent further drying in a vacuum dryer for 24 h before analysis. To introduce drug components, IBU (γCD:IBU was 1:1 molar ratio) was included in the solution prior to vapor diffusion.

##### Freeze-Drying Method

Based on preliminary studies and information found in the literature, and using the RA, three critical independent factors, namely the molar ratio of the API, incubation time, and percentage of the organic solvent, were selected. After determining the critical parameters, the factorial experimental design was used for the production. We chose the 3^3^ Box–Behnken experimental design, with which the effect of the 3 critical factors can be examined with a total of 15 experiments at 3 levels. Based on this, the produced samples are shown in Table 2.

After executing the Box–Behnken experimental design, we were able to calculate a second-order polynomial model, which resulted in a quadratic response surface. The response surface is based on the relationship between the specified CQA (dependent variable) and the chosen CMAs and CPPs (independent variables). The general form of the quadratic polynomial is described by Equation (1):(1)$$Y=\beta_{0}+\beta_{1}x_{1}+\beta_{2}x_{2}+\beta_{3}x_{3}+\beta_{12}x_{1}x_{2}+\beta_{13}x_{1}x_{3}+\beta_{23}x_{2}x_{3}+\beta_{11}x_{1}^{2}+\beta_{22}x_{2}^{2}+\beta_{33}x_{3}^{2}$$
where Y is the dependent variable, β_0_ is the axis section of the fitted function, *β*_1_, *β*_2_, and *β*_3_ are the linear coefficients, β_12_, β_13_, and β_23_ are the interaction coefficients between the three factors, β_11_, β_22_, and β_33_ are the quadratic coefficients. The response surface serves to understand the interactions of the main effects of 2 factors while keeping all others at a fixed level.

Switching from MeOH to EtOH was considered advantageous due to its lower toxicity [63] while still being a poor solvent for γCD-MOFs and initiates the nucleation of crystals [48], and at the same time it is an excellent solvent for encapsulating IBU [64]. Furthermore, regulatory guidelines, such as those outlined by the ICH, specify higher allowable organic solvent content for EtOH (5000 ppm) compared to MeOH (3000 ppm) [65]. To prevent crystal precipitation, it was crucial to determine the maximum quantity of EtOH that could be added to the solution. This was determined by a simple EtOH dosage, using the quantities also used in the vapor diffusion method. As a result, we found that an EtOH concentration of 22% leads to the precipitation of γCD-MOFs. To circumvent this, a maximum ethanol content of 20% was selected, enabling crystal formation from the solution during subsequent freeze-drying. The molar ratio of γ-CD/IBU was chosen based on the fact that the 1:1 CD/drug ratio is the most commonly used quantity in the literature. We were also curious about the effect of lack of and higher amounts of active ingredients. The incubation time was selected to be a suitable range also based on what we have found in the literature [48].

Freeze-drying was performed using the method generally used by the research group [66] with minor modifications. According to this, the freeze-drying processes of the samples were conducted using a laboratory apparatus Scanvac CoolSafe 100–9 Pro (LaboGeneApS, Lynge, Denmark) equipped with a Rotary Vane Vacuum Pump (Vacuubrand RZ 2.5, Wertheim, Germany). The device was pre-frozen to ensure that the 3-level sample trays positioned in the drying chamber were maintained at −40 °C. The process parameters were continuously monitored and recorded by the software (Scanlaf CTS16a02), which was detailed in Table 3. By the manufacturer’s recommendations, the sample vials were filled with a maximum volume of 1.5 mL of solutions. The primary drying phase was executed at −40 °C and 0.012 mbar for 16 h. Following this, secondary drying was performed at 0 °C and a pressure of 0.012 mbar for 2 h. Upon reaching stable product temperature and pressure conditions, the tray temperature was increased to room temperature (25 °C was set in the software). The samples were then dried at this temperature for an additional 4 h. When a stable constant pressure and product temperature were achieved, the samples were removed and stored at room temperature. The entire process lasted 30 h.

##### Morphological Examination of the Samples Produced by the Vapor Diffusion Method with Optical Microscopy

To confirm the morphological features of the vapor diffusion-produced samples, a Leica Q500MC imaging system (Leica Microsystems Cambridge Ltd., Cambridge, UK) was employed for room temperature examination with a cross-polarizer and a 4× magnification lens. We expected the cubic morphology known from the literature.

##### Morphological Examination of the Samples Produced by the Freeze-Drying Method with Scanning Electron Microscopy (SEM)

Freeze-dried samples were analyzed using scanning electron microscopy (Hitachi S4700, Hitachi Scientific Ltd., Tokyo, Japan). During the measurements, the following parameters were used: 10 mA amperage, 10 kV high voltage, and 1.3–13.1 mPa air pressure. Argon atmosphere was applied to make the sputter-coated samples conductive with gold-palladium. With this investigation method, we were curious about the conditions under which we can reproduce particles with the aforementioned cubic structure, and what other morphologies we can create with this technology.

##### Investigation of the Particle Size of the Formulations by Laser Diffraction

Malvern Mastersizer Scirocco 2000 (Malvern Instruments Ltd., Worcestershire, UK) was used to measure the particle size and particle size distribution of the samples. With the use of the dry dispersion apparatus, the solid particles were detected with 3.0 bar of dispersion air pressure and 75% vibration feed. Each measurement was repeated three times. The values of D[0.1], D[0.5], and D[0.9] (10%, 50%, and 90% of the volume distribution was below these values, respectively) were used to describe the particle size distribution (PSD). Span values were obtained from the PSD based on Equation (2).
(2)$$Span=\frac{D\left[0.9\right]-D[0.1]}{D[0.5]}$$

With the laser diffraction measurements, we were able to select the formulations with optimal size for pulmonary delivery, with criteria including a narrow particle size range and a diameter below 10 µm.

##### Examination of the Crystallinity of the Formulations by X-Ray Powder Diffraction (XRPD)

The prepared γCD-MOFs structure was assessed using a BRUKER D8 Advance X-ray powder diffractometer (Bruker AXS GmbH, Karlsruhe, Germany) equipped with Cu Kα radiation (λ = 1.5406 Å), a standard sample changer, and a VÅNTEC-1 line detector. The powder samples underwent scanning at 40 kV and 40 mA within an angular range of 3° to 40° 2θ, with a step time of 0.1 s and a step size of 0.01°. Using DIFFRACplus EVA software (version 5.2), we manipulated the acquired data (Kα2-stripping, background removal, and smoothing) and determined the samples’ crystallinity index percentage (%X_CI_) using the following Equation (3):(3)$${\%X}_{CI}=\frac{A_{crystalline}}{A_{crystalline}+A_{amorphous}}\times 100\%$$
where A_crystalline_ is the Area of all crystalline peaks, while A_amorphous_ is the Area of all amorphous peaks.

##### Examination of Thermal Properties with Differential Scanning Calorimetry (DSC)

DSC measurements were performed to investigate the thermal changes of IBU in the formulations. The thermal analysis was examined using the Mettler Toledo DSC 821e system and the STARe software V9.1 (Mettler Toledo Inc., Schwerzenbach, Switzerland). A continuous argon flow of 10 L·h^−1^ was used to heat about 2–3 mg of samples in sealed aluminum pans from 25 °C to 300 °C at a rate of 10 °C·min^−1^. We expected that the thermal stability of IBU would increase as a result of complexation.

##### Chemical Interaction Investigation with Fourier-Transformed Infrared Spectroscopy (FT-IR)

The interactions between γCD-MOFs and IBU were analyzed using AVATAR330 FT-IR spectrometer (Thermo Nicolet, Unicam Hungary Ltd., Budapest, Hungary). Samples were ground and compressed into pastilles at 10 t using 0.15 g of KBr. A total of 32 averaged scans were taken, and the samples were examined between 400 and 4000 cm^−1^ at an optical resolution of 4 cm^−1^. The collected spectra underwent standard adjustments, including averaging parallel measurements and applying baseline correction and peak normalization using Spectragryph-optical spectroscopy software v.1.2.16.1. [67].

##### Determination of IBU Concentration by High-Performance Liquid Chromatography (HPLC)

An Agilent 1260 HPLC (Agilent Technologies, Santa Clara, CA, USA) with a UV–VIS diode array detector was used to measure the concentration of IBU. A reversed-phase Kinetex^®^ EVO C18 column (5 mm, 150 mm × 4.6 mm (Phenomenex, Torrance, CA, USA)) and the column temperature was maintained at 25 °C. The mobile phases consisted of PBS phosphate buffer pH = 3.0 (A) and acetonitrile (B). Isocratic elution was carried out at a flow rate of 1 mL min^−1^ for 6 min with 45–55% A-B eluent composition. An amount of 10 mL of the samples was injected to determine the IBU concentration at 220 nm. The retention time of IBU was observed at 4.09 min. The IBU content was determined using a calibration curve in the concentration range of 4–500 µg mL^−1^. The correlation coefficient of the calibration curve was 0.9999 and the IBU showed good linearity in this interval.

##### In Vitro Aerodynamic Investigation

The aerosolization characteristics of the freeze-dried formulations were investigated using an Andersen Cascade Impactor (ACI) from Copley Scientific Ltd., Nottingham, UK. Figure 4. illustrates the modeling of the human respiratory system using the Andersen Cascade Impactor. A Critical Flow Controller Model TPK and a High-capacity Pump Model HCP5 were used to maintain a constant inhalation flow rate of 60 L/min. The actual flow rate was confirmed by a Flow Meter Model DFM 2000. During the measurement, an inhalation volume of 4 L was used, while the inhalation time was set to 4 s. The test employed a single-dose device which was Breezhaler^®^ from Novartis International AG, Basel, Switzerland, and size 3 Ezeeflo™ hydroxypropyl methylcellulose capsules from ACG-Associated Capsules Pvt. Ltd., Mumbai, India. Following inhalation, all parts of the equipment were thoroughly cleaned with ultra-pure water to remove any deposited IBU. At each measurement, a preparation containing 5 mg of IBU was investigated, which was adequately detectable in the measuring range of the HPLC. Quantification of the drug was achieved via HPLC method at a wavelength of 220 nm. Aerodynamic properties were analyzed using Inhalytix™ software v 2.0 from Copley Scientific Ltd., Nottingham, UK, which is specifically designed for such assessments. Fine Particle Fraction (FPF), Median Mass Aerodynamic Diameter (MMAD), and Emitted Dose (ED) were evaluated to provide a comprehensive understanding of the aerosolization process. The FPF is expressed as a percentage of the size fraction that is most likely to enter the lungs. This number shows the percentage of ED that could have a clinical effect. MMAD is the diameter that characterizes the average size of the inhaled aerosol. ED is the amount of API released from the inhaler. These parameters provide useful information regarding the aerodynamic characterization of dry powders; however, examination of the complete size distribution is very beneficial for accurate understanding. The optimal formulations were expected to bypass the upper airways and reach the lungs. For this purpose, in addition to the total distribution, the FPF% provides useful information. We wanted to increase the FPF% above 30% in order to approach or exceed the values published [33,34] so far with CD-MOFs carrier.

##### In Vitro Dissolution Measurement

Since there are no official regulatory requirements or protocols for the in vitro dissolution testing of inhalation products [68], we performed the dissolution according to the protocol previously published [9] by the research group. A modified methodology was utilized to investigate the release of IBU in vitro. An amount of 20 mL of artificial lung media with a pH of 7.4 ± 0.1 served as the dissolution medium at 37 °C. To guarantee adequate mixing, a 100 rpm magnetic stirrer was used. Amounts equal to the doses investigated with the Andersen Cascade Impactor were measured for the dissolution test. Aliquots of 1 mL of every sample were taken and collected at predetermined intervals of time (5, 10, 15, 30, 45, and 60 min). The samples were filtered using syringe filters having a pore size of 0.22 μm. For each sample, the release was repeated three times. Subsequently, quantification was performed through HPLC measurements. We expect to achieve rapid dissolution as a result of complexation.

##### Saturation Solubility

The solubility of IBU was characterized by the saturated solubility test using the most promising freeze-dried samples in addition to raw IBU. Briefly, an excess amount of the samples was dissolved in 5 mL of simulated lung fluid at 37 °C with constant stirring. After a few hours, the samples were filtered with a 0.22 μm syringe filter and concentrations were determined with HPLC method. All measurements were performed in triplicate for accuracy. This experiment aimed to demonstrate that the solubility of IBU was increased by the porous carrier.

##### Nuclear Magnetic Resonance (NMR) Spectroscopy

In order to decide whether our formulation appeared as an inclusion complex or just a solid dispersion, we performed an NMR analysis of the samples we considered applicable for pulmonary uptake. ^1^H NMR spectrum was recorded in D_2_O solution in a tube with a diameter of 5 mm. The instrument was a Bruker DRX-500 spectrometer with a 5 mm BBO Prodigy Probe (Bruker Biospin, Karlsruhe, Baden Württemberg, Germany) at 500 MHz.

##### Statistical Analysis

One-way ANOVA was used for statistical analysis, with Tukey’s test for further comparison. The results were presented as mean ± standard deviation (SD). A confidence threshold of *p* < 0.05 was used to establish significance between means. The effect on particle size diameters, Span values, and crystallinity indexes (%X_CI_) were investigated using Statistica^®^ 10.0.

## 3. Results and Discussion

### 3.1. Risk Assessment

During the risk assessment, we defined the CMAs and CPPs based on the literature and preliminary experiments, and subsequently assessed their impact on the CQAs. The estimated interactions between CQA and CPP/CMA are listed in Figure 5a. After classification, each attribute was assigned a unique severity value. The last step of the RA is the determination of the Risk Priority Value (RPN) of the CPP/CMAs, for which, in addition to the severity, we also evaluated the probability of occurrence of each factor using the previously applied 1-3-9 scale. The final result obtained during the risk assessment, namely the relationship between the relative occurrence and relative severity of the factors, is depicted in Figure 5b. Based on these evaluations, the most critical parameters affecting the quality of the final product are, in descending order of importance, production method > incubation time > molar ratio of CD and API > amount of organic solvent. The RA indicates that the production method is a key parameter in the successful preparation of nanoporous carriers. In our work, we used freeze-drying, due to the lack of literature data on this approach. The next three factors with the highest scores were the incubation time, molar ratio of CD and API, and the amount of organic solvent; therefore, the factorial experimental design (Table 2) was prepared for these factors based on the considerations discussed in the section Freeze-Drying Method.

With the obtained dependent variables, using the Statistica 10^®^ software, we can represent response surfaces by displaying two independent variables and one dependent variable, keeping the third dependent variable at a constant value. This is presented in detail in Section 3.2, Section 3.2.1 and Section 3.2.2.

### 3.2. Results of the Characterization of γCD-MOFs

#### 3.2.1. Investigation of Morphology and Particle Size

The samples produced by vapor diffusion showed regular cubic shape as described in the literature [32,33,34,35]. The micrographs of the materials mentioned before are shown in Figure 6, for samples without the API (a) and samples containing γCD:IBU in a 1:1 mol ratio (b).

Upon analyzing the freeze-dried samples depending on the different parameters, the SEM images (Figure 7a–d) showed particles with different shapes and surfaces. Due to the increasing concentration of ethanol, crystals with an increasingly smooth surface and angular crystals were obtained. The names of the samples shown in Figure 7 have been supplemented with the conditions (γ-CD/IBU; standing time; amount of EtOH, respectively) used for easier understanding and transparency. In the case of samples containing 20% and 10% ethanol, a spherical or angular morphology was characteristic, while those prepared with a volume ratio of 0% ethanol turned out to be rough, amorphous sheets.

The phenomenon can be explained by the interplay of several factors like freezing speed and solute concentration. Water, with the highest freezing point (0 °C), freezes the fastest among the solvents. This faster ice crystal formation can cause an amorphous state of the materials because they lack time to organize into a crystalline structure. Additionally, water’s ability to form hydrogen bonds [69] with the solute disrupts its orderly arrangement, leading to amorphous structures upon drying. In a 10% ethanol solution, the solute may be more evenly distributed within the ice, leading to uniform and spherical particles. In the 20% ethanol solution, the lower freezing point (−9 °C) of the organic solvent slows down the freezing time and allows the solute more time to crystallize, resulting in a more crystalline structure. In addition, since ethanol is a poor solvent, the γCD-MOFs enhanced the saturation of the solution, which also favors the formation of a crystalline structure.

The particle size of the freeze-dried samples was measured by laser diffraction. The d(0.1), d(0.5), d(0.9), and Span values obtained as the results of the measurement can be seen in Table 4. During further evaluation, the d(0.5) and Span values were our independent variables, depending on which we examined our dependent variables (Y value in Equation (1)) determined as CQA, which in our case were particle size and particle size distribution. With the obtained dependent variables, using the Statistica 10^®^ software, we could represent response surfaces by displaying two independent variables and one dependent variable, keeping the third dependent variable at a constant value (Figure 8a–d).

The main impacts and interaction effects of the independent variables were described by polynomial equations based on the results obtained. The significance of these variables and their interactions was assessed using analysis of variance (ANOVA). If the *p*-value for a factor was less than 0.05, it was found to have a significant effect. According to the analysis of the data, the relationship of the variables on particle size (d(0.5)) and distribution (Span) could be expressed with the Equations (4) and (5). The surface plots had regression coefficients of 0.9492 and 0.9999, and the adjusted R^2^ were 0.6458 and 0.9993, respectively.
(4)$$d\left(0.5\right)=12.546-4.276x_{1}+0.713x_{2}-1.333x_{3}-0.420x_{1}x_{2}-0.170x_{1}x_{3}+0.170x_{2}x_{3}+{0.382x}_{1}^{2}+{0.482x}_{2}^{2}-{1.449x}_{3}^{2}$$
(5)$$Span=3.472+1.840x_{1}+0.799x_{2}-1.095x_{3}+0.864x_{1}x_{2}-1.544x_{1}x_{3}-0.525x_{2}x_{3}-0.475x_{1}^{2}-0.285x_{2}^{2}-0.519x_{3}^{2}$$

The molar ratio of γCD and IBU had an effect on the particle size, as increasing the IBU concentration significantly decreased the particle size. The remaining two factors were not significant and therefore we did not take them into account in the analysis of the particle size. According to the response surface regression model, the value of the dependent variable was always represented as a function of two independent variables, keeping the third variable at a constant, medium level value. Figure 8a represented well the effect of molar ratio on particle size. We demonstrated that by keeping the EtOH% value at 0 level (10%), the standing time did not affect the change in particle size in any direction, while when examining the molar ratio of the two components, it is clear that by increasing the amount of IBU, the particle size decreased linearly. Interestingly, the opposite of this phenomenon was described by Rajaram et al. [43] while they observed an increase in particle size after the complexation of adenosine with βCD-MOFs and explained it by the manufacturing process. Our assumption about this appearance is that the pores in the cubic crystals [37] can provide extra space outside the cyclodextrin cavity for IBU molecules to wedge in, which does not necessarily affect the morphology of the cubic particle, but can inhibit crystal growth.

The change in the Span values was impacted by each of the three parameters. The most significant effect here was also the change in the γCD/IBU ratio. The factor had a positive effect on the Span, i.e., increasing the amount of IBU also linearly increased the size distribution of the particles. Both the linear and quadratic effects of all three factors were significant, and the influence of the factors on each other indicated a significant change both linearly and quadratic. Overall, we were able to establish the magnitude of the effects of the factors on the Span value in the following order: γCD/IBU molar ratio > EtOH% > standing time. Since the interpretation of the effects is more complex than the former sequence, Figure 8b–d provide assistance in the interpretation, where we used a two-dimensional response surface to illustrate the directions of the Span value changes with the changes of the components. The observed changes are due to alterations in particle size and crystallinity, which consequently influenced the distribution of particle size distributions in this manner.

Using laser diffraction measurements, we successfully identified the formulations with the optimal particle size for pulmonary delivery which was discussed in more detail in Section 3.3.

#### 3.2.2. Crystallinity of γCD-MOFs Prepared with Different Parameters

The XRPD pattern of the γCD-MOF sample produced by conventional vapor diffusion (Figure 9) had characteristic crystalline peaks at 4.2°, 5.8°, 9.2°, 12.3°, 16.3°, and 23.0°. This coincided with what was described in the literature [39,41,42]. The same reflections are not visible in the raw IBU and γCD, which indicates that a completely new crystal structure was obtained during crystallization. Our samples produced by freeze-drying were compared with the diffractogram of the sample produced by vapor diffusion. The above-mentioned reflections also appear in freeze-dried samples at 4.1°, 5.8°, 9.0°, 13.4°, 16.7°, and 23.2°. The similarity of the peaks with the control sample confirms that the structure of the γCD-MOFs was successfully formed with and without active ingredient even in one step with the freeze-drying method. The degrees of crystallinity calculated by the software are displayed in Table 5 and their representation depending on the various factors is shown in Figure 10a–c. The relationship between the variables related to crystallinity could be expressed by Equation (6) with R^2^ = 0.9992 (R^2^) and adj. R^2^ = 0.9944 (R^2^) which indicated a great correlation.
(6)$$\%X_{CI}=66.608-7.983x_{1}-0.442x_{2}+23.058x_{3}-1.750x_{1}x_{2}-6.550x_{1}x_{3}-4.975x_{2}x_{3}-{4.660x}_{1}^{2}-{2.973x}_{2}^{2}+{0.977x}_{3}^{2}$$

The concentration of organic solvent had the greatest influence; the explanation was the same as that described in Section 3.2.1. Standing time and amount of IBU also had a detectable effect on crystallinity. According to the strength of the effects on the change in crystallinity, the following order was established: EtOH% > γCD/IBU molar ratio > standing time.

#### 3.2.3. Thermal Behavior of the Formulations

Although the XRPD results strongly indicated the formation of the γCD-MOF structure and the incorporation of the IBU, we also investigated the formation of the complexes with DSC measurements in order to firmly support our assumptions. In Figure 11, the DSC curve of raw IBU revealed two distinct endothermic peaks at 78.4 °C and 282.5° which were related to the melting point and degradation of the API. The thermodynamic properties of IBU showed distinct changes after encapsulating into γCD-MOFs, because the endothermic peak belonging to IBU disappeared due to the incorporation in the γCD cavity or into the crystal lattice of the carrier [70]. The γCD-MOF samples all showed a broad endothermic peak around 79.2 °C, corresponding to the loss of water and moisture [71]. As indicated by the results of DSC measurements, the thermal stability of IBU increased following complexation, in accordance with our expectations.

#### 3.2.4. Secondary Interactions between IBU and γCD-MOF

The intermolecular interactions between γCD-MOFs and IBU were investigated using FT-IR. The γCD-MOFs samples exhibit the same characteristic peaks as raw γCD at 3526 cm^−1^, 2928 cm^−1^, and 1158 cm^−1^ (Figure 12) suggesting that the structure of γCD in the organometallic systems has not changed [72]. Wide absorption peaks between 3700 and 3000 cm^−1^ were exhibited by all γCD-MOFs, which belonged to the –OH stretching vibrations of γCD. In samples which contained IBU, the minimums of the characteristic peaks were shifted towards shorter wavelengths which is presented in Table 6. The results indicated an interaction between the -OH groups of γCD-MOFs and IBU, as others have reported similar phenomena [71]. Among the samples, the most intense effect of these shifts was observed in the spectra of Samples 3 and 4, indicating that these samples formed the most hydrogen bonds between IBU and the carrier. Characteristic absorptions [73] for raw IBU were found at 2958 cm^−1^, 1720 cm^−1^, 1509 m^−1^, 1460 cm^−1^, 1228 cm^−1^, 1182, and 1068 cm^−1^. The spectra of the IBU-containing samples showed some characteristic absorptions of IBU, proving that IBU was included in the formulation [64]; however, not all characteristic peaks appeared, for which different explanations can be found in the literature. On the one hand, the absence of characteristic peaks may be due to the fact that the IBU completely encapsulated in the carrier [71], but on the other hand, the absence of peaks could also be caused by the fact that the complex contained a small percentage of IBU compared to the carrier [64]. However, the presence of IBU is certain, and its interaction with the carrier, in addition to the shift of the previously mentioned –OH bands, is absent from all complex spectra of the characteristic peak with the highest intensity at 1720 cm^−1^ (stretching vibration of the carbonyl group (C=O stretching)) together with the peak at 1230 cm^−1^ (stretching vibration of the carbonyl group (C–O stretching)) of raw IBU. New minima appeared in the complex around 1570 cm^−1^, which are more intense with higher active ingredient content (Samples 3, 4, 7, and 8). This possibly refers to the anionized form of IBU (–COO^−^) and this peak is assigned to the carboxylate anion. We assume a H-bonding interaction between the carboxylate anion and γCD [64] where IBU acts as a proton acceptor. In summary, the results suggest that there was an interaction between γCD-MOFs and IBU while the –OH groups on the glucose units can form hydrogen bonds with the lone pairs of electrons through the carboxyl groups of IBU.

### 3.3. In Vitro Aerodynamic Investigation

Samples between 1 and 10 μm d(0.5) value were chosen to be measured for lung deposition in vitro. Based on Table 4, these samples were with numbers 3, 4, 8, 10, and 15. This size range was established based on the following information: particles’ average diameters must be in a narrow range to effectively transfer the API to the lung [31,74]. Furthermore, particles with diameters between 1 and 5 μm are usually deposited in the deep lung regions [75] while those larger than 10 μm are usually deposited in the upper respiratory tract [31,71]. The particles smaller than 1 μm are exhaled [31]. Nevertheless, studies have shown that deposition is not only affected by particle size. For example, Chvatal et al. were the first who reported the improvement of the deposition of large porous meloxicam particles in an aqueous medium [76]. Based on their research, particles larger than 5 µm showed better aerodynamic properties compared to particles with smaller diameters due to the porous structure. The samples mentioned above had very different morphologies and internal structural arrangements. Our assumption was that although the cubic structure may have great aerodynamic properties due to its homogeneous nanoporous structure, the spherical shape with a wrinkled [77,78] surface is also favorable for achieving deep lung deposition, especially with the diameters and Span values measured for Samples 3 and 4. After the experiments, our assumption was confirmed, as these two samples achieved a significant difference compared to the other formulations at stages 3 and 4 modeling the bronchial part (Figure 13.). Almost the entire amount of IBU from Samples 8, 10, and 15 were deposited in the part modeling the upper airways; therefore, they were not suitable for further investigations for the purpose of developing a dry powder inhalation formulation. At stage 3 and 4, Samples 3 and 4 showed a significant difference compared to the other samples at *p* < 0.01 significance level, and in relation to each other we could also show a significant difference at *p* < 0.05 significance level. The FPF values of Samples 3 and 4 were 38.10 ± 5.06% and 47.18 ± 4.18%, while their MMAD values were 5.06 ± 0.21 μm and 4.15 ± 0.57 μm, respectively. For the pursuit of transparency, the parameters characterizing the aerodynamic properties are summarized in Table 7. These FPF values proved to be excellent compared to several studies. Y. Zhou et al. [33] achieved a 33.12% FPF for γCD-MOF/D-Limonene, in the case of γCD-MOF/Paeonol, H. Li et al. [34] reported 17.85% FPF (maximum 27.73% with additional aerosolization-increasing excipients), and Y. Huang et al. [79] demonstrated that with γCD-MOF/Cyclosporine A, the best FPF result was 39.69% by using polyethylene glycol 10,000 as a modulator. Compared to these published values, we achieved similar (Sample 3) or better (Sample 4) FPF results, without the use of other aerosolization aids or modulators. The two samples were structurally and morphologically identical, differing only in the size of the particles. The acquired results can be explained by the fact that γCD-MOFs aerosolization was mostly dependent on particle size rather than flowability, as previously reported by Zhou et al. [33,39]. Zhou et al. reported in 2020 that the particle size of CD-MOFs is critical to the successful deposition, and in 2021, using additional aerosolization-promoting excipients, they came to the same conclusion. With this work, we demonstrated that, in terms of efficient deposition, the particle size will determine whether or not CD-MOFs have a completely homogenous nanoporous structure. In addition, we achieved results that exceeded the FPF 30% target value established at the beginning of the experiment.

### 3.4. In Vitro Dissolution

The bioavailability of drugs intended for pulmonary administration is highly dependent on the solubility and dissolution rate of the drug particles deposited in the lung. The in vitro drug dissolution from the reference raw IBU and γCD-MOFs formulations with the greatest MMAD and FPF values (Samples 3 and 4) was investigated by modeling lung conditions. The results of the tests are presented in Figure 14. While the untreated raw IBU reached the maximal concentration after 30 min, the active ingredient was completely dissolved from the formulations during the first 5 min, i.e., we achieved a very fast dissolution by incorporating it into the γCD-MOFs and thereby improving the solubility of IBU. In comparison, only 28.7 ± 4.4% of the raw IBU was dissolved during the first 5 min. The increased solubility of IBU was discussed further in Section 3.5. In addition to complexation, some other factors may have played a role in the rapid release of IBU from the complex. It has already been reported that the spray-dried semi-crystalline γCD-MOF/Ketoconazole [48] exhibited the best dissolution properties compared to the amorphous-like and highly crystalline γCD-MOF/Ketoconazole; therefore, the reduction in crystallinity observed in our samples may have facilitated the phenomenon of rapid dissolution. Furthermore, Y. Zhou reported the influence of wettability on the enhanced dissolution rate [32], indicating that the increase in wettability was a significant mechanism in this process, which can also be attributed to complexation. According to the study, the mechanism was explained as the destruction of the CD-MOFs structure when the complex came into contact with water. It was also mentioned that the small particle size and large surface area of particles could improve the dissolution rate of API, but it has already been proven that the decrease in crystallinity had a greater effect [48]. The spray-dried semi-crystalline γCD-MOF/Ketoconazole exhibited the best dissolution properties compared to the amorphous-like and highly crystalline γCD-MOF/Ketoconazole; thus, we believe that the speed of dissolution could be caused by the formation of the semi-crystalline structure in addition to complexation, not by changes in particle size or surface area. In summary, our dissolution tests demonstrated that γCD-MOFs could efficiently improve the dissolution rate of IBU under lung conditions which is a very important point in the treatment of CF because the slow dissolution of IBU is a limiting step for penetration through the mucous membrane [80,81].

### 3.5. Saturation Solubility

It is a well-known fact that the solubility of IBU is pH-dependent [82]; therefore, it is not surprising that IBU dissolves better under a certain concentration in lung fluid (pH = 7.4) compared to water. In order to demonstrate the increased solubility of IBU in artificial lung fluid after the complexation with γCD-MOFs, we performed a saturated solubility test of the samples applied in Section 3.4. The results are presented in Table 8.

Based on the results, the solubility of IBU was increased by more than 10 times compared to raw IBU even in this alkaline medium. It is already well known from the literature that γCD-MOFs can dramatically improve the solubility of varied poorly water-soluble molecules due to the complexation. We discussed in the previous section the possible mechanisms of the increase in fast dissolution, but we are sure that the increased solubility mainly plays a role, which happened due to complexation. The increased solubility ensures better therapeutic applicability in pulmonary application, especially in the treatment of CF [83].

### 3.6. Verification of the Inclusion Complex Structure

Structural characterizations revealed that our formulations with great aerodynamic properties (Sample 3 and Sample 4) had a lower crystallinity and a spherical shape compared to the complexes produced by the traditional vapor diffusion method. To the best of our knowledge, until now only Hartlieb et al. [84] published NMR spectrum about this topic with potassium salt of IBU. They used the conventional vapor diffusion method with γCD and formed γCD-MOF/IBU-potassium cocrystals. However, with using the salt form of IBU, the IBU anions could not be located in the pores of cyclodextrins. In order to prove the complexation of IBU within the porous frameworks, an NMR study was performed with the mentioned samples. The NMR spectrum of Sample 3 is presented in Figure 15. The numbering of the protons of both γCD and IBU are shown on the spectrum. Spectra of raw materials were recorded as references. The shifts of the detected chemical shifts positions on the horizontal axis (induced shift, Δδ) were defined as the differences between the chemical shifts of the raw materials and complexes (Δδ = Δδfree − Δδcomplex). The results are presented in Table 9. The induced shifts were similar in both Sample 3 and Sample 4. Based on the literature, the observed differences in displacements suggest that IBU was located within the pores of γCD [85,86]. Furthermore, we found that our formulation did not contain any residual organic solvent.

## 4. Conclusions

In this research, instead of using the commonly used vapor diffusion method, γCD-MOFs were created by freeze-drying, which has not been published yet in the literature. We were the first to implement the QbD method in the production of inhalable γCD-MOFs. By defining the CMAs and CPPs, with the help of the RA, we successfully fulfilled our expectations for the QTPP, which means, we successfully produced preparations that could potentially be used in the treatment of CF. Based on QbD, we identified three critical factors, namely the molar ratio of the IBU to the γCD, incubation time, and the percentage of the organic solvent. Morphology, particle size, particle size distribution, and crystallinity could be tuned by changing the factors. The molar ratio of γCD and IBU had an effect on the particle size, as increasing the IBU concentration significantly decreased the particle size. Our assumption about this appearance was that the pores in the cubic crystals can provide additional space outside of the γCD cavity for IBU molecules to wedge in, which do not necessarily affect the morphology, but can inhibit crystal growth. This made it possible to increase the IBU content in the γCD-MOFs, as the higher IBU content created a more favorable size for pulmonary intake. In the in vitro aerodynamic study, the most favorable results were observed in samples with a 1:2 molar ratio of γCD to IBU when the initial solution contained 10% organic solvent. The MMAD[μm] and FPF[%] values of the sample incubated for 48 h were 4.15 ± 0.57 and 47.18 ± 4.18, respectively, while the values of the immediately frozen sample were 5.06 ± 0.21 and 38.10 ± 5.06. We demonstrated that, in terms of deposition efficiency, particle size is a key factor. From the aforementioned samples, nearly the entire quantity of IBU dissolved within the first 5 min during the in vitro dissolution tests; therefore, we achieved a rapid dissolution of the active ingredient for the effective treatment of CF. The effectiveness was further enhanced by the fact that the solubility of IBU was drastically increased due to the complexation. In order to prove that an inclusion complex was indeed formed, we supplemented our XRPD, DSC, and FT-IR structural analysis methods with NMR spectroscopy, which adequately supported our assumptions. Our formulations could be suggested for further investigations, while γCD-MOFs are expected to be promising carriers for IBU delivery by pulmonary route.

## Figures and Tables

**Figure 1 pharmaceutics-16-01361-f001:**
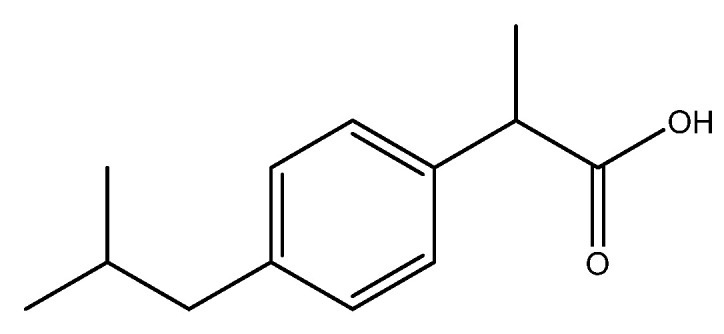
The chemical structure of IBU.

**Figure 2 pharmaceutics-16-01361-f002:**
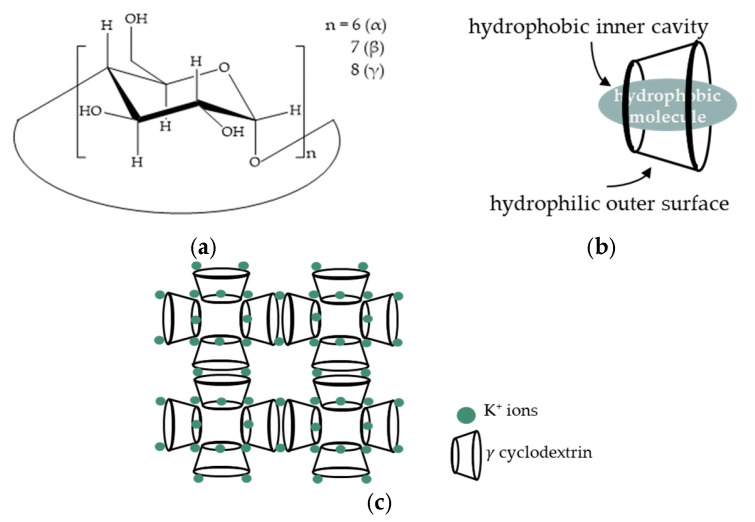
The chemical compositions of α-, β-, and γ-cyclodextrins (**a**), along with a schematic representation of CDs (**b**) and CD-MOFs (**c**).

**Figure 3 pharmaceutics-16-01361-f003:**
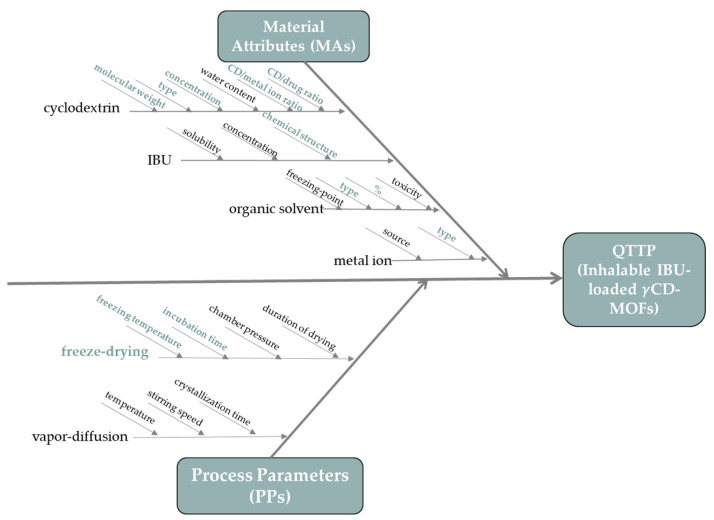
Ishikawa cause-effect diagram for IBU-loaded CD-MOFs.

**Figure 4 pharmaceutics-16-01361-f004:**
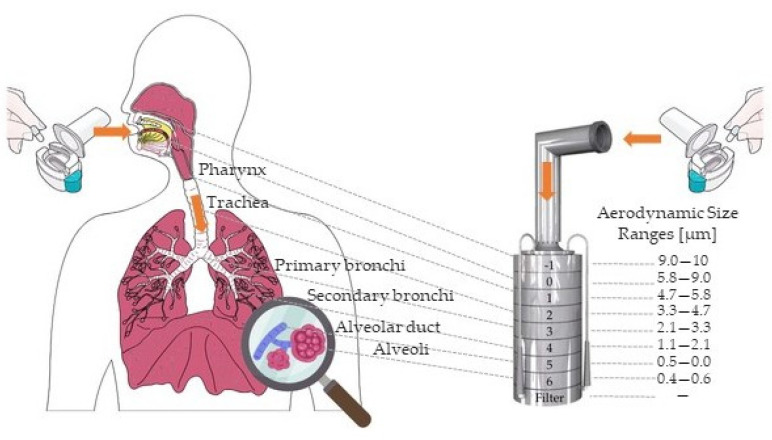
Schematic representation of the working principle of Andersen Cascade Impactor.

**Figure 5 pharmaceutics-16-01361-f005:**
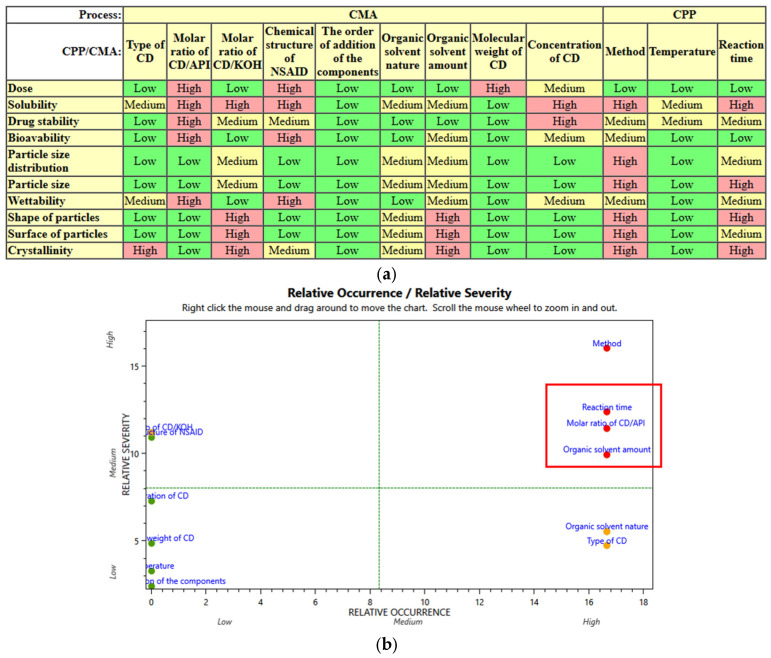
Evaluation of correlation between CQAs-CPPs/CMAs (**a**) based on the preliminary RA, and the relative occurrence and relative severity of the factors (**b**).

**Figure 6 pharmaceutics-16-01361-f006:**
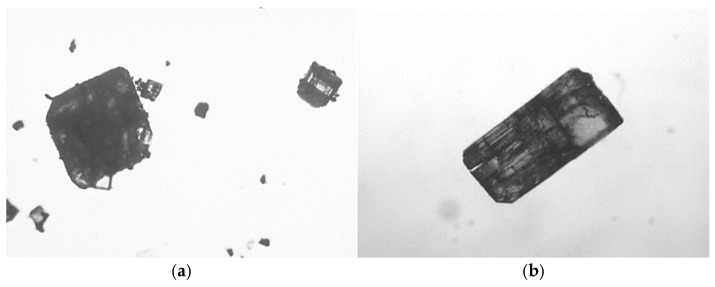
Light microscopic pictures of samples prepared by vapor diffusion method: γCD-MOFs without (**a**) and in the presence of IBU (**b**).

**Figure 7 pharmaceutics-16-01361-f007:**
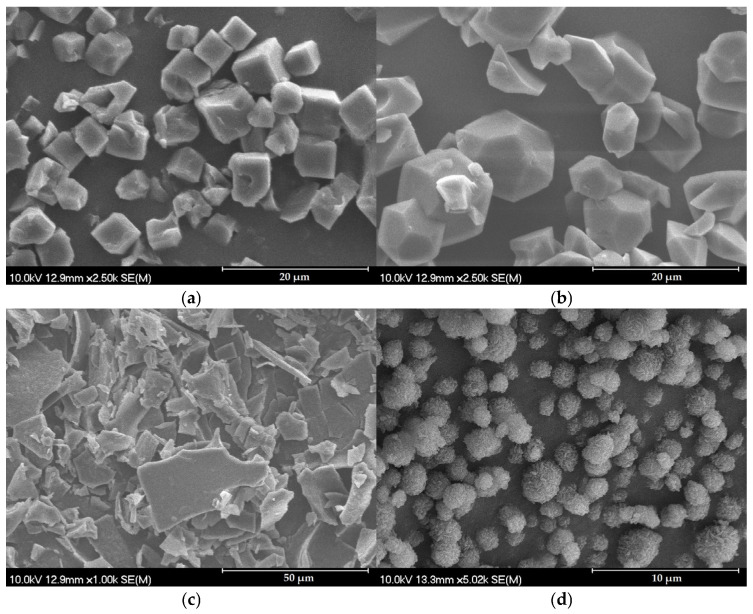
SEM micrographs of freeze-dried Sample 10 (1:1 mol; 0 h; 20%) (**a**), 13 (1:1 mol; 24 h; 10%) (**b**), 11 (1:1 mol; 48 h; 0 h) (**c**), and 4 (1:2 mol; 48 h; 10%) (**d**).

**Figure 8 pharmaceutics-16-01361-f008:**
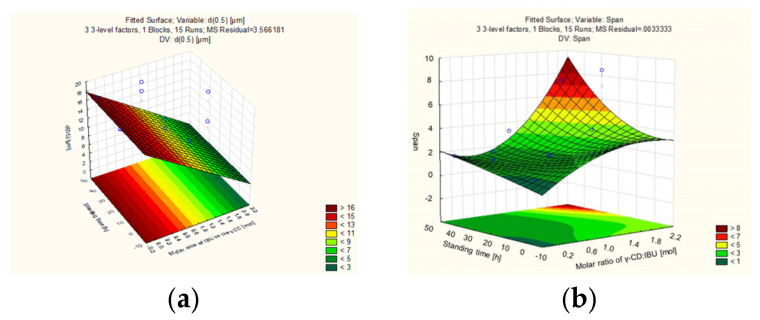
Three-dimensional surface plots of the effect of independent variables on the particle size (**a**) and Span values (**b**–**d**).

**Figure 9 pharmaceutics-16-01361-f009:**
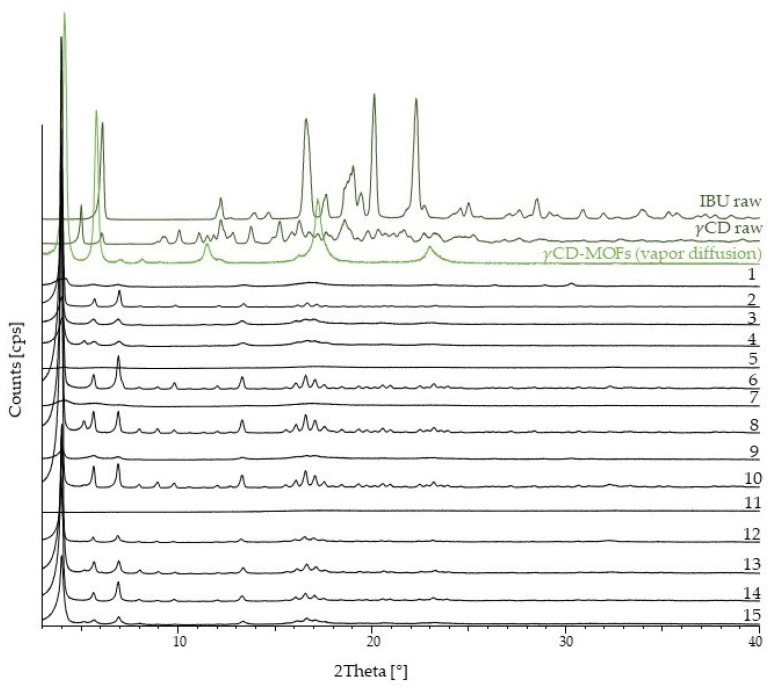
XRPD patterns of raw materials and γCD-MOFs with IBU prepared by vapor diffusion and freeze-dried method.

**Figure 10 pharmaceutics-16-01361-f010:**
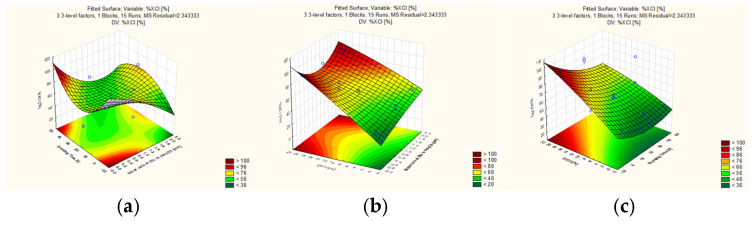
Three-dimensional surface plots of the effect of independent variables on the crystallinity indexes (**a**–**c**).

**Figure 11 pharmaceutics-16-01361-f011:**
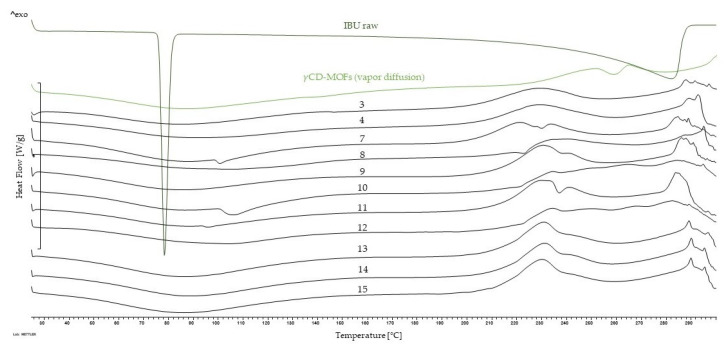
DSC curves of raw IBU and γCD-MOFs containing IBU.

**Figure 12 pharmaceutics-16-01361-f012:**
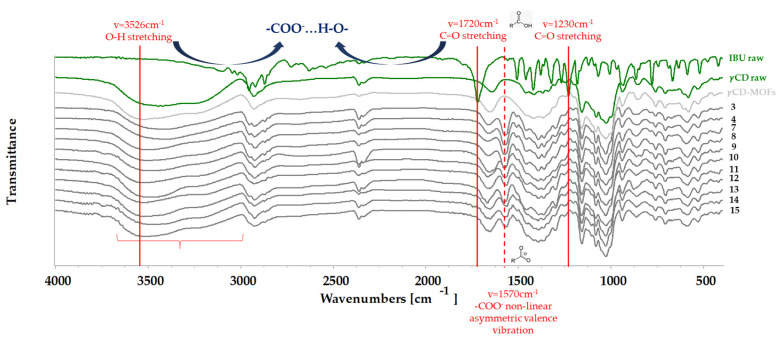
FT-IR spectra of control γCD-MOFs and freeze-dried γCD-MOFs containing IBU.

**Figure 13 pharmaceutics-16-01361-f013:**
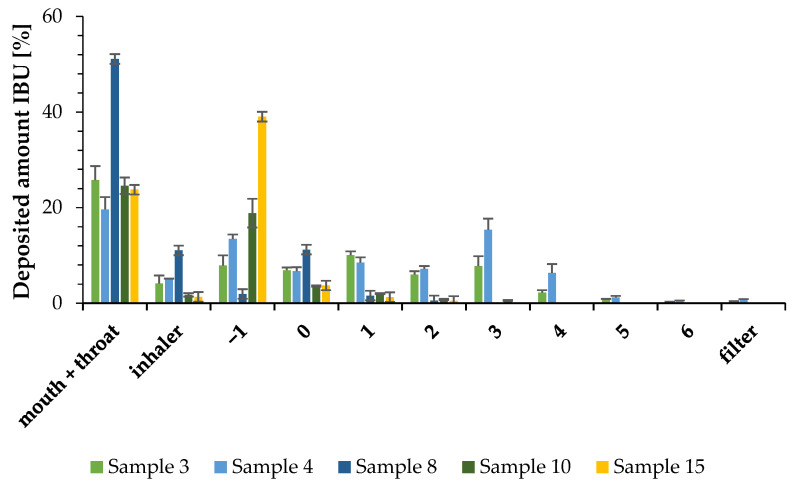
In vitro aerodynamic distribution of the freeze-dried samples.

**Figure 14 pharmaceutics-16-01361-f014:**
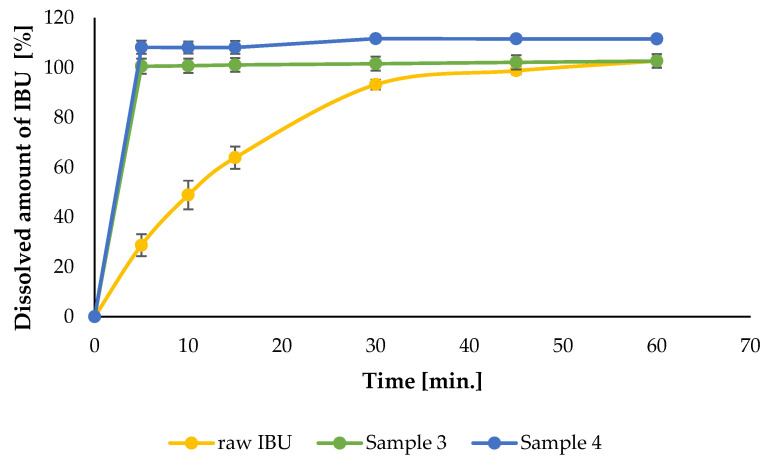
Release profile of raw IBU and CD-MOFs with the best MMAD and FPF% values in simulated lung media at 37 °C.

**Figure 15 pharmaceutics-16-01361-f015:**
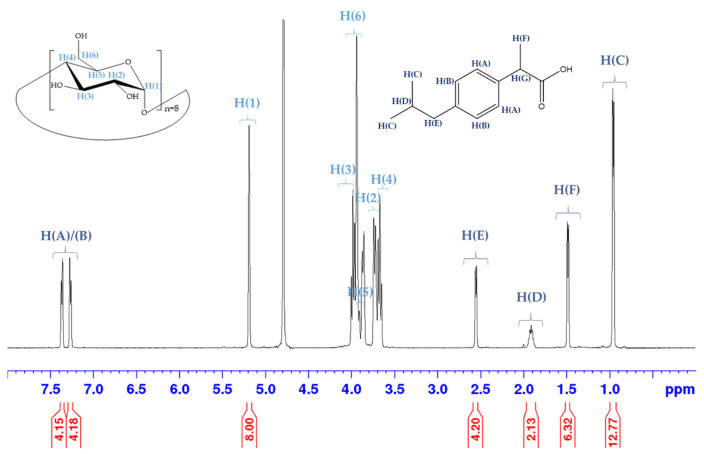
^1^H NMR spectrum of Sample 3 in D_2_O.

**Table 1 pharmaceutics-16-01361-t001:** The elements of the QTPP, the target values for the elements, and the justification of the target values.

QTPP Element	Target	Justification
Indication	Treatment of CF, reducing inflammation and pain	While inflammation is a characteristic of the lung disease of CF and contributes to lung destruction, high-dose IBU therapy could igh-dose IBUon should be ility. a be helpful in slowing disease progression [16,17,18].
Target patients	Children who suffer from CF genetic diseases	CF is a rare genetic disorder; it is especially necessary to treat it in childhood [12,13,14].
Route of administration	Pulmonary	Inhalation administration could be more effective in the treatment of CF; it requires a lower dose of the API and reduces adverse effects.
Dosage form	Powder	DPIs are environmentally friendly, provide high stability, and are easy to use [58].
Targeted organ part for the clinical effect	Entire lung	Inflammation and bronchiectasis are more common in the upper lobe, while mucus plugging and air trapping are more frequent in the lower lobes [59,60]. Therefore, it is essential to target the entire lung, for which appropriate aerodynamic properties are important for proper utilization [31] ouldd.
Dissolution profile	Fast dissolution	It is important to ensure that a high dose of API quickly reaches the lungs and for a rapid disintegration of the final product to happen [61,62] to achieve maximum therapeutic levels in the blood stream.
Primary packaging	Capsule	DPIs can be easily administered from the capsule while ensuring equal dosing and protection from humidity.

**Table 2 pharmaceutics-16-01361-t002:** Freeze-dried samples are made using the Box–Behnken factorial experimental design.

Run Order	Molar Ratio of γ-CD:IBU	Standing Time [h]	EtOH [%]
1	1:0	0	10
2	1:0	48	10
3	1:2	0	10
4	1:2	48	10
5	1:0	24	0
6	1:0	24	20
7	1:2	24	0
8	1:2	24	20
9	1:1	0	0
10	1:1	0	20
11	1:1	48	0
12	1:1	48	20
13	1:1	24	10
14	1:1	24	10
15	1:1	24	10

**Table 3 pharmaceutics-16-01361-t003:** Freeze-drying parameters used to prepare the samples are listed in Table 2.

Process	Process [h]	P_chamber_ [hPa]	T_product_ [°C]	T_shelf_ [°C]
Device pre-freezing	0–4	609.242–594.295	-	+22 to −40
Sample freezing	4–8	594.295–565.493	23 to −40	−29 to −40
Primary drying	8–24	0.012	−46 to −38	−40
Secondary drying I	24–26	0.012	−38 to −16	−40 to 0
Secondary drying II	26–30	0.012	−16 to +7	0 to +26

**Table 4 pharmaceutics-16-01361-t004:** d(0.1), d(0.5), d(0.9), and Span values based on laser diffraction measurements. Data means ± SD (n = 3 independent measurements).

Sample	d(0.1) [μm]	d(0.5) [μm]	d(0.9) [μm]	Span
1	6.6 ± 0.4	15.2 ± 0.3	34.9 ± 6.2	0.9 ± 0.4
2	6.7 ± 0.5	16.2 ± 0.3	33.4 ± 1.9	1.6 ± 0.4
3	1.9 ± 0.1	3.9 ± 0.1	14.0 ± 3.8	3.2 ± 0.9
4	1.2 ± 0.2	3.2 ± 0.3	25.3 ± 5.9	7.4 ± 1.4
5	6.6 ± 2.3	14.7 ± 2.6	25.8 ± 8.8	2.0 ± 0.6
6	6.6 ± 2.0	14.8 ± 5.3	34.3 ± 11.9	1.9 ± 0.2
7	4.8 ± 0.4	16.8 ± 0.9	151.0 ± 95.4	8.7 ± 4.4
8	4.3 ± 0.3	10.1 ± 1.0	22.5 ± 2.3	1.8 ± 0.2
9	4.3 ± 0.2	14.3 ± 0.4	38.5 ± 2.0	2.4 ± 0.1
10	4.1 ± 0.0	9.5 ± 0.7	46.7 ± 7.6	4.5 ± 0.5
11	4.2 ± 0.2	14.9 ± 1.3	53.6 ± 1.9	3.3 ± 1.2
12	6.5 ± 0.2	16.9 ± 1.1	62.4 ± 3.2	3.3 ± 1.7
13	6.0 ± 0.2	13.8 ± 0.2	30.3 ± 6.7	1.8 ± 0.5
14	5.1 ± 0.7	11.7 ± 0.8	24.7 ± 7.2	1.7 ± 0.4
15	4.2 ± 0.6	8.6 ± 1.1	19.5 ± 0.9	1.8 ± 0.1

**Table 5 pharmaceutics-16-01361-t005:** Crystallinity degrees of freeze-dried samples.

Sample	%X_CI_ [%]
1	82.9
2	92.3
3	57.2
4	59.6
5	27.3
6	92.4
7	50.9
8	89.8
9	46.8
10	91.1
11	42.3
12	66.7
13	56.9
14	59.5
15	56.8

**Table 6 pharmaceutics-16-01361-t006:** -OH stretching vibration positions on the spectra.

Sample Name	Detected Peak [cm^−1^]
CD-MOFs control sample	3526.0
Sample 3	3418.8
Sample 4	3417.0
Sample 7	3502.8
Sample 8	3504.4
Sample 9	3503.9
Sample 10	3522.1
Sample 11	3503.3
Sample 12	3524.0
Sample 13	3524.1
Sample 14	3480.1
Sample 15	3524.4

**Table 7 pharmaceutics-16-01361-t007:** Summary of the characteristics of the dry powders with the best aerodynamic properties. Data are means ± SD (n = 3 independent measurements).

Sample Name	MMAD [μm]	FPF [%]	EF [%]
Sample 3	5.06 ± 0.21	38.10 ± 5.06	72.34 ± 1.05
Sample 4	4.15 ± 0.57	47.18 ± 4.18%	84.87 ± 0.08

**Table 8 pharmaceutics-16-01361-t008:** Solubility of the freeze-dried samples and raw IBU in simulated lung media (pH = 7.4 ± 0.1) at 37 °C.

Sample Name	C_IBU_ [mg/mL]
Sample 3	56.9 ± 5.5
Sample 4	53.9 ± 3.4
raw IBU	4.5 ± 2.3

**Table 9 pharmaceutics-16-01361-t009:** ^1^H chemical shifts of γCD and IBU protons in the inclusion complexes (Sample 3 and Sample 4) and raw materials.

γCD Proton	δ_free_ [ppm]	δ_Sample 3_ [ppm]	Δδ [ppm]	δ_Sample 4_ [ppm]	Δδ [ppm]
H(1)	5.2105	5.1890	0.0215	5.1894	0.0211
H(2)	3.7461	3.7355	0.0106	3.7357	0.0105
H(3)	4.0327	3.9876	0.0451	3.9886	0.0441
H(4)	3.6882	3.6735	0.0147	3.6737	0.0145
H(5)	3.9497	x	x	x	x
H(6)	3.9839	3.9291	0.0548	3.9292	0.0547
IBU Proton	δ_free_ [ppm]	δ_Sample 3_ [ppm]	Δδ [ppm]	δ_Sample 4_ [ppm]	Δδ [ppm]
H(A/B)	7.3486	7.3146	0.0339	7.3145	0.0341
H(E)	2.5792	2.5574	0.0219	2.5578	0.0214
H(D)	1.9393	1.9236	0.0157	1.9186	0.0206
H(F)	1.5068	1.4885	0.0182	1.4893	0.0175
H(C)	0.9695	0.9611	0.0085	0.9622	0.0073

## Data Availability

No new data were created or analyzed in this study. Data sharing is not applicable to this article.

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
