# Peer review of "Preparation of Ibuprofen-Loaded Inhalable γCD-MOFs by Freeze-Drying Using the QbD Approach"

_pharmaceutics, 2024, doi:10.3390/pharmaceutics16111361_

Round 1

Reviewer 1 Report

Comments and Suggestions for Authors

The article presents a novel and valuable approach for preparing ibuprofen-loaded γ-cyclodextrin metal-organic frameworks (γCD-MOFs) using freeze-drying, supported by the Quality by Design (QbD) framework. However, several areas require significant revisions, and some scientific suggestions can strengthen the findings.

Points for Revisions:

-The freeze-drying method described for the γCD-MOF preparation lacks sufficient technical details. Specific conditions for freezing (temperature, freezing speed) should be elaborated on. Also, the rationale behind selecting freeze-drying over other methods, such as spray drying, is briefly mentioned but should be compared regarding physicochemical outcomes and scalability.

- Characterizing γCD-MOFs would benefit from a more profound structural analysis, especially regarding the frameworks' porosity and influence on drug loading. Techniques such as Brunauer–Emmett–Teller (BET) surface area measurements could be considered to quantify the porosity and further verify the nanoporous structure, strengthening the argument for γCD-MOFs as carriers.

-The dissolution studies show promising fast-release profiles, but a more detailed discussion of the mechanisms driving the enhanced solubility and release should be included. Specifically, the roles of particle size reduction, surface area enhancement, and crystallinity changes should be explored in more detail with reference to the molecular interactions observed in the FT-IR and NMR studies.

-The paper would benefit from directly comparing γCD-MOFs and other pulmonary drug delivery carriers, such as aerogel. e.g. In the study conducted by Khodov et al. (https://doi.org/10.3390/ijms24086882), the conformational equilibrium of mefenamic acid was explored using NMR techniques, revealing how the chemical environment of silica aerogels influenced the equilibrium between different conformers. Similarly, ibuprofen, like mefenamic acid, exhibits high conformational dynamics (https://doi.org/10.1016/j.molliq.2020.113113), which may also play a significant role in the release dynamics of ibuprofen from cyclodextrin-based metal-organic frameworks (MOFs) (https://doi.org/10.1016/j.supflu.2017.11.009) . In the context of your research, it would be valuable to briefly discuss the advantages and disadvantages of freeze-drying (sublimation drying) and supercritical fluids. Furthermore, the conformational lability of ibuprofen could influence its behavior during such processes, potentially affecting its release profile and bioavailability in drug delivery systems.

-Although DSC and FT-IR confirm thermal stability, more must be discussed about the formulations' long-term stability. It is crucial to address how the freeze-dried γCD-MOFs maintain their structure and performance over time under storage conditions, especially for inhalable formulations.

The study presents a pioneering approach to pulmonary drug delivery that has the potential to significantly impact cystic fibrosis treatment. With major revisions focusing on improved methodological clarity and deeper analysis, the paper can offer robust contributions to the scientific community.

Author Response

Reply to Referee comments

For Reviewer 1

Thank you very much for your remarks. We greatly appreciate your advices. Here you

can see listed all of the modifications made in the paper according to your suggestions

(shown in green colour in the text).

-The freeze-drying method described for the γCD-MOF preparation lacks sufficient technical details. Specific conditions for freezing (temperature, freezing speed) should be elaborated on. Also, the rationale behind selecting freeze-drying over other methods, such as spray drying, is briefly mentioned but should be compared regarding physicochemical outcomes and scalability.

Thank you for your observation. We recognize the importance of providing a more detailed explanation of the freeze-drying method. In the revised version, we would like to incorporate the following information into the manuscript:

„Freeze-drying was performed using the method generally used by the research group [62] with minor modifications. According to this, the freeze-drying processes of the samples were conducted using a laboratory apparatus Scanvac CoolSafe 100–9 Pro (LaboGeneApS, Lynge, Denmark) equipped with a Rotary Vane Vacuum Pump (Vacuubrand RZ 2.5, Wertheim, Germany).  The device was pre-frozen to ensure that the 3-level sample trays positioned in the drying chamber were maintained at -40 °C. The process parameters were continuously monitored and recorded by a software (Scanlaf CTS16a02), which was detailed in Table 3. In accordance with the manufacturer's recommendations, the sample vials were filled with a maximum volume of 1.5 mL of solutions. The primary drying phase was executed at −40 °C and 0.012 mbar for 16 hours. Following this, secondary drying was performed at 0 °C and a pressure of 0.012 mbar for 2 hours. Upon reaching stable product temperature and pressure conditions, the tray temperature was increased to room temperature (25 °C was set in the software). The samples were then dried in this temperature for an additional 4 hours. When a stable constant pressure and product temperature was achieved, the samples were removed and stored at room temperature. The entire process lasted 30 hours.”

„Table 3. Freeze-drying parameters used to prepare the samples listed in Table 2.”

Process

Process [h]

Pchamber [hPa]

Tproduct [°C]

Tshelf [°C]

Device pre-freezing

0-4

609.242-594.295

-

+22 to -40

Sample freezing

4-8

594.295-565.493

23 to -40

-29 to -40

Primary drying

8-24

0.012

-46 to -38

-40

Secondary drying I

24-26

0.012

-38 to -16

-40 to 0

Secondary drying II

26-30

0.012

-16 to +7

0 to +26

Since we have added a table to the manuscript, we have changed the numbering of all subsequent tables.

„Vapor-diffusion is the most widely used method for synthesizing CD-MOFs, howev-er, this process can take 3-5 days or weeks, requires the use of a large amount of organic solvent and provides low product yield [40]. Furthermore, in several cases the active in-gredient was loaded in a separate step [41–43], which mean a multi-step process. Conse-quently, the development of alternative protocols for CD-MOF production was seemed es-sential, with proposed advantages including enhanced product yield, reduced prepara-tion time, and minimized amount of organic solvent. Alternative production techniques already exist in the literature [38], such as microwave- ultrasound- or modulator-assisted synthesis. The use of modulators further complicated the production, as they were often inseparable from the final product [44]. In addition, the previously mentioned techniques are required a subsequent drying process. Despite the freeze-drying method occasional mention in literature as a final step of synthesis [42,45–47], there has been limited comprehensive investigation into its suitability for complex preparation from solutions, even though analogous experiments have been carried out using the spray-drying method [36,48]. In comparison to the traditional vapor-diffusion method, spray-drying presents a promising alternative for production, offering advantages such as higher product yield, reduced synthesis time, and tunable particle properties, all achieved in a single step without the need for a subsequent drying process. However, the technique has certain lim-itations, for instance, its application is not advantageous for heat-sensitive, highly viscous, or highly foaming materials. The alternative freeze-drying technique can provide a solu-tion to the problem. In comparison to spray-drying, freeze-drying is a more time-consuming, complex, and expensive process. However, it retains the advantages as-sociated with spray drying over traditional vapor-diffusion method. Moreover, freeze-drying can address specific challenges that arise during spray drying and provides an opportunity to the production of a stable product. Taking these factors into considera-tion, we strongly believe that developing a freeze-drying production process of CD-MOFs could represent a valuable strategy.”

- Characterizing γCD-MOFs would benefit from a more profound structural analysis, especially regarding the frameworks' porosity and influence on drug loading. Techniques such as Brunauer–Emmett–Teller (BET) surface area measurements could be considered to quantify the porosity and further verify the nanoporous structure, strengthening the argument for γCD-MOFs as carriers.

Thank you for your suggestion. Unfortunately, our institute currently lacks access to a BET surface area measurement technique. However, we intend to explore the loading of the active ingredient and the enhancement of its content in future studies related to the preparations discussed in the publication.

-The dissolution studies show promising fast-release profiles, but a more detailed discussion of the mechanisms driving the enhanced solubility and release should be included. Specifically, the roles of particle size reduction, surface area enhancement, and crystallinity changes should be explored in more detail with reference to the molecular interactions observed in the FT-IR and NMR studies.

Thank you for the comment. Following the advice, we modified the manuscript in three places (3.4. In Vitro Dissolution; 3.5. Saturation solubility and 4. Conclusions):

„The increased solubility of IBU was discussed further in chapter 3.5. Saturation solubility. In addition to complexation, some other factors may have played a role in the rapid re-lease of IBU from the complex. It has already been reported that the spray-dried semi-crystalline ?CD-MOF/Ketoconazole [48] exhibited the best dissolution properties compared to the amorphous-like and highly crystalline ?CD-MOF/Ketoconazole therefore the reduction in crystallinity observed in our samples may have facilitated the phenome-non of rapid dissolution. Furthermore, Y. Zhou reported the influence of wettability on the enhanced dissolution rate [32], indicating that the increase in wettability was a significant mechanism in this process, which can also be attributed to complexation. According to the study, the mechanism was explained as the destruction of the CD-MOFs structure when the complex came into contact with water. It was also mentioned that the small par-ticle size and large surface area of particles could improve the dissolution rate of API, but it has already been proven that the decrease in crystallinity had a greater effect [48]. The spray-dried semi-crystalline ?CD-MOF/Ketoconazole exhibited the best dissolution prop-erties compared to the amorphous-like and highly crystalline ?CD-MOF/Ketoconazole, thus, we believe that the speed of dissolution could be caused by the formation of the semi-crystalline structure in addition to complexation, not by changes in particle size or surface area.”

„It is already well known from the literature that ?CD-MOFs can dramatically improve the solubility of varied poorly water soluble molecules due to the complexation. We discussed in the previous chapter the possible mechanisms of the increase in fast dissolution, but we are sure that the increased solubility mainly plays a role, which happened due to complexation.”

The effectiveness was further enhanced by the fact that the solubility of IBU was drastically increased due to the complexation.”

-The paper would benefit from directly comparing γCD-MOFs and other pulmonary drug delivery carriers, such as aerogel. e.g. In the study conducted by Khodov et al. (https://doi.org/10.3390/ijms24086882), the conformational equilibrium of mefenamic acid was explored using NMR techniques, revealing how the chemical environment of silica aerogels influenced the equilibrium between different conformers. Similarly, ibuprofen, like mefenamic acid, exhibits high conformational dynamics (https://doi.org/10.1016/j.molliq.2020.113113), which may also play a significant role in the release dynamics of ibuprofen from cyclodextrin-based metal-organic frameworks (MOFs) (https://doi.org/10.1016/j.supflu.2017.11.009) . In the context of your research, it would be valuable to briefly discuss the advantages and disadvantages of freeze-drying (sublimation drying) and supercritical fluids. Furthermore, the conformational lability of ibuprofen could influence its behavior during such processes, potentially affecting its release profile and bioavailability in drug delivery systems.

Thank you for your comment. In this publication, we evaluated the performance of our carriers in comparison to other γCD-MOF carriers, as detailed in the In Vitro Aerodynamic Investigation section. We compared our work with results published by Y. Zhou et al. (doi:10.1016/j.ijpharm.2021.120825; doi:10.1016/j.apsb.2020.07.018), H. Li et al. (doi:10.1016/j.ijpharm.2020.119649), and Y. Huang et al (doi:10.1016/j.biopha.2024.116174). We acknowledge the importance of comparing our γCD-MOFs with other carriers. However, as our research group is currently working on experiments with various CD-based carrier types, we intend to conduct these comparisons at a later stage, integrating them with our own results for a more comprehensive analysis. This approach will ensure a more robust evaluation of performance across different carriers. We sincerely appreciate the publication recommendations. The papers authored by Khodov et al. discussed complex NMR techniques and analyses, which we do not possess the expertise to compare with our measurements. Additionally, in the first publication aerogels were identified as „optimal carriers for transdermal drug delivery” but our focus is primarily on pulmonary intake systems. Consequently, we believe that the comparison presented in this publication may not be directly applicable to our research context. The NMR testing methods appear both interesting and valuable, and we hope to have the opportunity in the future to conduct similarly high quality investigations on the conformers. The SASD technique also presents a promising alternative for processing thermolabile APIs, however, our knowledge of this specific technique is limited. Moreover, we did not find any literature about the use of this technique for producing CD-MOFs, suggesting it is worth exploring further.         
We appreciate your suggestion to emphasize the advantages of freeze-drying method. We have made the necessary modifications in the manuscript to highlight this point.    
We also agree that the conformation of ibuprofen can influence its behavior in formulations. We believe that using CDs with ibuprofen is beneficial, as CDs are known as stability enchancers. Additionally, we consider that the gentle freeze-drying method is also advantageous for formulating ibuprofen which has a low melting point. We will further explore this topic in our upcoming research.

-Although DSC and FT-IR confirm thermal stability, more must be discussed about the formulations' long-term stability. It is crucial to address how the freeze-dried γCD-MOFs maintain their structure and performance over time under storage conditions, especially for inhalable formulations.

Thank you for your comment. We acknowledge the significance of conducting long-term stability tests, which are essential for assessing formulation performance. While we currently have limited data on this aspect, we recognize its importance and have included plans for the optimization of formulations and the investigation of their long-term stability in our future research. The present study focused on freeze-drying as a potential CD-MOF synthesis method, and we aimed to demonstrate an important area of ​​use for the products. The optimization of the preparation and the investigation of long-term stability are considered as a later goal.

Reviewer 2 Report

Comments and Suggestions for Authors

In this manuscript, γCD-MOF composed of γCD and K+ ions were utilized to form complex with the model drug IBU. The authors implement the QbD method in the production of inhalable γCD-MOFs. Based on QbD, three critical factors were identified as the molar ratio of the IBU to the γCD, incubation time, and the percentage of the organic solvent. The as-synthesized inclusion complex were proved by XRPD, DSC and FT-IR structural analysis methods with NMR spectroscopy. The γCD-MOFs were expected to be promising carriers for IBU delivery by pulmonary route. The manuscript provides thorough study and adequate experiment details, it is also well organized. I’d suggest accepting it with the following issues resolved:
1.  Why was the freeze-frying method used for γCD-MOFs, “which hasn’t been published in the literature” doesn’t sounds convincing, the authors should specify the benefit of freeze-drying compare to traditional vapor diffusion crystallization method, more background information should be added to the intro part.

2.  According to Figure 9, the XRPD patterns of γCD-MOFs samples using vapor diffusion method and freeze-drying method are not aligned well, any explanation for that?

3. Three main factors were investigated including molar ratio of the IBU to the γCD, incubation time, and the percentage of the organic solvent, have the authors considered other factors such as the freeze-drying temperature?

4. In table 2, why 15 experiments at 3 levels were shown instead of 33  (a total of 27 experiments), what are the other possibilities being eliminated?

5. The Aerodynamic Investigation on γCD-MOFs was not discussed.

6. It would be helpful to label EtOH% in Figure 7 to compare the organic solvent percentage influence, instead of just calling sample 10, 13, 11 and 4.

Author Response

Reply to Referee comments

For Reviewer 2

Thank you very much for your remarks. We greatly appreciate your advices. Here you

can see listed all of the modifications made in the paper according to your suggestions

(shown in purple colour in the text).

  1. Why was the freeze-frying method used for γCD-MOFs, “which hasn’t been published in the literature” doesn’t sounds convincing, the authors should specify the benefit of freeze-drying compare to traditional vapor diffusion crystallization method, more background information should be added to the intro part.

Thank you for this valuable observation. We have completed the manuscript with an additional part that discusses the reasons underlying freeze-drying, based on the recommendations provided. Since Reviewer 1 had similar comment, the modification was marked in green:

„Vapor-diffusion is the most widely used method for synthesizing CD-MOFs, howev-er, this process can take 3-5 days or weeks, requires the use of a large amount of organic solvent and provides low product yield [40]. Furthermore, in several cases the active in-gredient was loaded in a separate step [41–43], which mean a multi-step process. Conse-quently, the development of alternative protocols for CD-MOF production was seemed es-sential, with proposed advantages including enhanced product yield, reduced prepara-tion time, and minimized amount of organic solvent. Alternative production techniques already exist in the literature [38], such as microwave- ultrasound- or modulator-assisted synthesis. The use of modulators further complicated the production, as they were often inseparable from the final product [44]. In addition, the previously mentioned techniques are required a subsequent drying process. Despite the freeze-drying method occasional mention in literature as a final step of synthesis [42,45–47], there has been limited comprehensive investigation into its suitability for complex preparation from solutions, even though analogous experiments have been carried out using the spray-drying method [36,48]. In comparison to the traditional vapor-diffusion method, spray-drying presents a promising alternative for production, offering advantages such as higher product yield, reduced synthesis time, and tunable particle properties, all achieved in a single step without the need for a subsequent drying process. However, the technique has certain lim-itations, for instance, its application is not advantageous for heat-sensitive, highly viscous, or highly foaming materials. The alternative freeze-drying technique can provide a solu-tion to the problem. In comparison to spray-drying, freeze-drying is a more time-consuming, complex, and expensive process. However, it retains the advantages as-sociated with spray drying over traditional vapor-diffusion method. Moreover, freeze-drying can address specific challenges that arise during spray drying and provides an opportunity to the production of a stable product. Taking these factors into considera-tion, we strongly believe that developing a freeze-drying production process of CD-MOFs could represent a valuable strategy.”

  1. According to Figure 9, the XRPD patterns of γCD-MOFs samples using vapor diffusion method and freeze-drying method are not aligned well, any explanation for that?

Thank you for your question. A similar phenomenon can also be seen in the literature (https://doi.org/10.1016/j.ultsonch.2022.106003; https://doi.org/10.1021/acsabm.3c00162; https://doi.org/10.1016/j.foodchem.2016.06.013). The loading of the API, particle size along with the particle size distribution can influence the overall diffraction pattern, potentially complicating the subsequent analysis. We made a comparison of the XRPD diffractograms of our samples with those documented in the literature and successfully identifying the primary reflections noted at XRPD analysis (The main characteristic reflections of γCD-MOFs are marked on the XRPD diagram in the following publication: https://doi.org/10.1016/j.foodchem.2020.127839). Since we were able to detect the characteristic main reflections in both samples, we conclude that the minor differences can be attributed to variations in the loading of the active ingredient, particle size, and particle size distribution resulting from the two different production methods which included different conditions and solvents.

  1. Three main factors were investigated including molar ratio of the IBU to the γCD, incubation time, and the percentage of the organic solvent, have the authors considered other factors such as the freeze-drying temperature?

In this study, we employed a protocol commonly used by our research group for freeze-drying, with minor modifications developed during preliminary experiments. The factors were selected to determine whether we could produce γCD-MOF crystals (under controlled technical conditions), as described in the literature. This was the most critical aspect of the study, as freeze-drying, like spray-drying, often leads to amorphous products and our goal was to achieve crystallization using a lyophilizer. We successfully ended the research and and we revealed the possibility to produce γCD-MOFs with freeze-drying both with and without the active ingredient while varying the molar ratio of the IBU to the γCD, incubation time, and the percentage of the organic solvent. We are sure that parameters such as temperature, temperature dynamics, and drying time influence the production of these formulations since even the shape of the storage container and the measured volume of the solution can affect the process. Unfortunately, due to the large number of parameters, we cannot examine all of them together. This is the reason why we chose the three parameters mentioned in the article as the three most critical factors, which were established during the Risk Assessment. Our primary plan for the future is to optimize the formulation and investigate the encapsulation efficiency. We also aim to explore the effects of temperature, cooling rate, and drying time, and we plan to reproduce the production of carriers or formulations using different types of lyophilizers.

  1. In table 2, why 15 experiments at 3 levels were shown instead of 33  (a total of 27 experiments), what are the other possibilities being eliminated?

Thank you for the question. We have attached a Table for easier understanding. The entire 33 plans would have been realized with 27 experiments based on the Table below. Instead, in our research we made only the samples marked with x from the table, this is the essence of the Box-Behnken simplification.

Table: The 33 Box-Behnken factorial experimental design (each "x" in the figure represents one experiment.The central product is produced three times to obtain information about the reliability of the process).

If we examine the effects of three factors at three levels, it typically requires a full factorial design of 33 experiments which results 27 experiments. However, this can be reduced to fewer experiments while still obtaining reliable results. One such simplification method is the Box-Behnken design, which allows for the investigation of the influence of 3 critical factors at 3 levels using only 15 experiments. If it becomes necessary to perform any experiments that were not carried out, we can obtain information from the quadratic response surfaces generated by the Box-Behnken design results.

  1. The Aerodynamic Investigation on γCD-MOFs was not discussed.

Thank you for your comment. In this publication, we evaluated the performance of our carriers in comparison to other γCD-MOF carriers, as detailed in the In Vitro Aerodynamic Investigation section. We compared our work with results published by Y. Zhou et al. (doi:10.1016/j.ijpharm.2021.120825; doi:10.1016/j.apsb.2020.07.018), H. Li et al. (doi:10.1016/j.ijpharm.2020.119649), and Y. Huang et al (doi:10.1016/j.biopha.2024.116174). We only investigated those formulations with the Andersen Cascade Impactor, whose D(0.5) value was below 10 μm, because the larger particle size is definitely not suitable for pulmonary intake. The discussion was only carried out for those formulations that showed assessable results. Samples 8, 10 and 15 were not discussed after the in vitro aerodynamic measurements, as they were seemed unsuitable for pulmonary delivery. The results for these samples could not be evaluated by the Inhalytix™software.

The following section is considered as discussion of 3.3. From the In Vitro Aerodynamic investigation chapter:

„These FPF values proved to be excelent compared to several studies. Y. Zhou et al. [33] achieved a 33.12% FPF for γCD-MOF/D-Limonene, in the case of γCD-MOF/Paeonol, H. Li et al. [34] reported 17.85% FPF (maximum 27.73% with additional aerosolization-increasing excipients), and Y. Huang et al. [76] demonstrated that with γCD-MOF/Cyclosporine A, the best FPF result was 39.69% by using polyethylene glycol 10000 as modulator. Compared to these published values we achieved similar (Sample 3) or better (Sample 4) FPF results, without the use of other aerosolization aids or modulators. The two samples were structurally and morphologically identical, diffe-ing only in the size of the particles. The acquired results can be explained by the fact that γCD-MOFs aerosolization was mostly dependent on particle size rather than flowability, as previously reported by Zhou et al [33,39]. Zhou et al. reported in 2020 that the particle size of CD-MOFs is critical to the successful deposition, and in 2021, using additional aerosolization-promoting excipients, they came to the same conclusion. With this work, we demonstrated that, in terms of efficient deposition, the particle size will determine whether or not CD-MOFs have a completely homogenous nanoporous structure. In addition, we achieved results that exceeded the FPF 30% target value established at the beginning of the experiment.”

Furthermore, in the conclusion section, we mentioned the following regarding the In Vitro Aerodynamic investigation:

„The MMAD[μm] és FPF[%] values ​​of the sample incubated for 48 hours were 4.15±0.57 and 47.18±4.18 respectively, while the values ​​of the immediately frozen sample were 5.06±0.21 and 38.10±5.06. We demonstrated that, in terms of deposition efficiency, particle size is a key factor.”

  1. It would be helpful to label EtOH% in Figure 7 to compare the organic solvent percentage influence, instead of just calling sample 10, 13, 11 and 4.

Thank you for your comment. We understand that it is difficult to interpret the evaluation due to the large number of samples and variable parameters. Although EtOH% had the greatest effect on the change in morphology, Sample 13 and 4 also contained 10% EtOH, but one showed angular and the other showed spherical morphology. For this reason, instead of adding only the amount of EtOH, we tried to indicate all parameters, since the amount of ibuprofen also had an influence. Required changes are marked in purple.

Reviewer 3 Report

Comments and Suggestions for Authors

This manuscript focuses on the development of ibuprofen-loaded inhalable γ-cyclodextrin metal-organic frameworks (γCD-MOFs) using a freeze-drying method guided by a Quality by Design (QbD) approach. The manuscript demonstrates that the formulations have favourable aerodynamic properties for pulmonary delivery, significantly improving ibuprofen solubility and dissolution rates, which could potentially be used for treating cystic fibrosis.

The authors should be commended for a well-researched and thorough covering both the methodology and characterisation of ibuprofen-loaded γCD-MOFs for pulmonary drug delivery. It demonstrates a structured approach, particularly through the use of the QbD methodology, and including detailed experimental results supported by various characterization techniques such as SEM, DSC, XRPD, and NMR.

Again while I have stated that the quality of English does not limit my understanding of the research, the manuscript requires a thorough edit to remove numerous grammatical and typographical errors. It does prove an annoyance to see simple errors remain after review by multiple co-authors.

Line 25 - The acronym FPF is introduced without definition or stating the relevance.

Line 328 - intervall

Line 394 - Assesment

Line 501 - Manufactoring

Line 609 - Chvalal et al.

Line 682- IBUeven (space missing)

Line 727 - és

Author Response

Reply to Referee comments

For Reviewer 3

Again while I have stated that the quality of English does not limit my understanding of the research, the manuscript requires a thorough edit to remove numerous grammatical and typographical errors. It does prove an annoyance to see simple errors remain after review by multiple co-authors.

Line 25 - The acronym FPF is introduced without definition or stating the relevance.

Line 328 - intervall

Line 394 - Assesment

Line 501 - Manufactoring

Line 609 - Chvalal et al.

Line 682- IBUeven (space missing)

Line 727 – és

Thank you for your comment, we have corrected the manuscript in the necessary places. (shown in orange colour in the text).
